# Efficacy and Safety of Esaxerenone for Essential Hypertension: A Systematic Review and Meta-Analysis of Randomized Controlled Trials [note 1]

**DOI:** 10.3390/jcm14165663

**Published:** 2025-08-11

**Authors:** Abdelrahman Hafez, Ahmed Abdelaziz, Ahmed Mansour, Ibrahim Kamal, Ali Bakr, Ahmed Farid Gadelmawla, Hanaa Elsayed, Mohamed Reyad Mohamed, Karim Ali, Mohamed Elhelw

**Affiliations:** 1Medical Research Group of Egypt (MRGE), Negida Academy, Arlington, MA 02474, USA; abdelrahmanhafez13498@gmail.com (A.H.); ahmedmansourama2000@gmail.com (A.M.); ibrahem.kamoo98@gmail.com (I.K.); dralibakr93@gmail.com (A.B.); ahmedsaid140000@gmail.com (A.F.G.); hanaa.elsayed525@gmail.com (H.E.); 2Cardiovascular Medicine Department, Mayo Clinic, Phoenix, AZ 85054, USA; 3Division of Cardiology, Montefiore Health System, Albert Einstein College of Medicine, Bronx, NY 10461, USA; 4Faculty of Medicine, Al-Azhar University, Cairo 11884, Egypt; 5Faculty of Medicine, Menoufia University, Menoufia 32511, Egypt; 6Faculty of Medicine, Zagazig University, Zagazig 44519, Egypt; 7Cardiology Department, Banner University Medical Center, Phoenix, AZ 85006, USA; mohamed.reyad1@bannerhealth.com; 8Department of Internal Medicine, Hennepin County Medical Center, Minneapolis, MN 55415, USA; kareemmmedical@gmail.com; 9Department of Internal Medicine, University of Kentucky Healthcare, University of Kentucky, KY 40536, USA; mel242@uky.edu

**Keywords:** essential hypertension, esaxerenone, HTN usual care, meta-analysis

## Abstract

**Background/Objectives:** Esaxerenone, a novel non-steroidal mineralocorticoid receptor antagonist, has shown promising results in the treatment of essential hypertension (HTN). This study aims to comprehensively analyze the effectiveness of esaxerenone to control BP in patients with essential HTN. **Methods:** A systematic search was performed on PubMed, Scopus, Web of Science (WOS), and Cochrane Library from inception until January 2025 for randomized controlled trials (RCTs) comparing esaxerenone with standard HTN usual care. The primary outcome of interest was mean changes in sitting systolic (SBP) and diastolic blood pressure (DBP). Other secondary outcomes were mean changes in 24 h-SBP and 24 h-DBP, and target BP achievement. Safety outcomes, such as adverse events and increased levels of potassium/uric acid, were also assessed. **Results:** Our meta-analysis included four studies with a total of 1981 patients, all conducted in Japan. Esaxerenone demonstrated dose-dependent blood pressure reductions. At 5 mg, sitting DBP decreased by 4.22 mmHg (95% CI −8.72 to 0.29; *p* = 0.07) and SBP by 9.07 mmHg (95% CI −17.69 to −0.45; *p* = 0.04), while the 1.25 mg dose showed smaller, non-significant reductions. For 24 h measurements, only the 5 mg dose achieved significant reductions. Target BP achievement increased by 48% overall (RR 1.48, 95% CI 1.11 to 1.96; *p* = 0.007), with stronger effects at 5 mg (RR 2.05, 95% CI 1.03 to 4.08; *p* = 0.04). Esaxerenone significantly reduced nocturnal SBP by 10.62 mmHg (95% CI −14.01 to −7.23; *p* < 0.001). **Conclusions:** In patients with essential HTN, esaxerenone was shown to be safe and well-tolerated compared with usual care. Long-term data on safety is warranted.

## 1. Introduction

Resistant hypertension (HTN), defined as uncontrolled blood pressure (BP) despite proper management with at least three antihypertensive drugs of varied classes, is a substantial risk factor for major cardiovascular events [1,2,3]. The prevalence of resistant hypertension has been reported to be in the range of 20% to 35% [3]. Large-scale studies, including data from the All of Us Research Program encompassing over 81,000 participants, demonstrate that apparent treatment-resistant hypertension affects approximately 14.4% of hypertensive individuals in the United States [4].

The pathophysiology underlying resistant hypertension involves complex, interconnected mechanisms primarily centered on dysregulation of the renin–angiotensin–aldosterone system (RAAS) [5]. Aldosterone exerts its effects through mineralocorticoid receptors (MRs) in renal distal tubular cells, promoting sodium reabsorption and potassium excretion, thereby expanding blood volume and elevating BP. Thus, higher aldosterone levels are linked to a greater risk and severity of hypertension [6,7]. Consequently, pharmacological agents that inhibit the action of aldosterone on MRs exhibit great therapeutic efficacy. Established MR antagonists, including spironolactone and eplerenone, have been demonstrated as therapeutic agents for managing treatment-resistant HTN [8,9]. However, the administration of these drugs is linked with various treatment-related side effects, particularly those that are sex hormone-related [10].

Esaxerenone, a newly developed non-steroidal MR blocker taken orally, has been recognized for its ability to lower BP, of which recent phase 1 and 2 clinical trials revealed its effectiveness in BP reduction and favorable tolerability [11,12,13].

Numerous studies have yielded conflicting results regarding the efficacy and safety profile in treating hypertension [13,14,15,16]. Additionally, the previous systematic review was based solely on evidence from two randomized controlled trials (RCTs) and lacked comprehensive assessment, subgroup analyses, and outcome evaluations [17]. Thus, we conducted a systematic review and meta-analysis of RCTs to assess the efficacy and safety of esaxerenone in adult patients with essential hypertension.

## 2. Materials and Methods

We followed the proposed guidelines in the Cochrane Handbook for Systematic Reviews of Interventions and also the Preferred Reporting Items for Systematic Reviews and Meta-analyses (PRISMA) guidelines while conducting this systematic review and meta-analysis [18,19]. Before the start of the search, a review protocol was entered into the PROSPERO database (ID: CRD420251048639).

### 2.1. Literature Search

A thorough literature search was performed across PubMed, Scopus, WOS, and Cochrane Library from inception to January 2025, using the following search terms: (Esaxerenone) AND (Hypertension). A detailed search strategy according to each database is shown in Appendix A. Additionally, reference lists of eligible articles were screened for additional citations to ensure a comprehensive citation analysis.

### 2.2. Eligibility Criteria

Two independent reviewers (MM and AM) screened retrieved references in a two-step process: title and abstract screening for citation eligibility, and full-text screening to identify randomized controlled trials (RCTs) that met the eligibility criteria.

We included only RCTs enrolling patients with essential HTN and reported outcomes of interest in patients allocated to esaxerenone compared to usual care. The studies had to assess the primary outcome of interest, mean change in sitting SBP (SBP of 140 to <180 mmHg). Additional secondary outcomes were the mean change in sitting diastolic BP (DBP) of 90 to <110 mmHg, 24 h ambulatory BP of ≥130/80 mmHg, and a nighttime SBP measured ≥120 mmHg during the follow-up. The risk of adverse events, including the potassium/uric acid ratio, was also assessed.

We further excluded studies whose patients were diagnosed with secondary hypertension or malignant hypertension, animal studies, data from conference meetings, or unpublished data.

### 2.3. Risk of Bias Assessment

Two authors (AH and IK) independently evaluated the quality of the included RCTs using the Cochrane Risk of Bias assessment tool 2 (ROB2) (Cochrane, London, UK), assessing five bias domains: randomization process, intervention adherence, missing data handling, outcome measurement, and selective reporting. Each domain was rated as “low risk”, “some concerns”, or “high risk”, contributing to an overall study rating [20]. Any discrepancies between the authors were resolved through discussion or by a third author.

In line with Cochrane guidelines, which recommend against assessing publication bias when fewer than ten studies are included, we did not conduct a formal publication bias assessment due to the limited number of eligible studies [19].

### 2.4. Data Extraction and Analysis

Data from included studies were extracted and recorded in a standardized Excel sheet. The extracted data were the last author, publication year, sample size, location, participant demographics, follow-up duration, hypertension grade, comorbidities, baseline blood pressure, inclusion criteria, conclusion, and primary outcomes.

A standard two-stage pairwise meta-analysis was performed. Continuous data were expressed as mean, standard deviation (SD), and the total patients. For each continuous outcome, the pooled mean difference (MD) with its 95% confidence interval (CI) was calculated using the Der Simonian–Laird random-effect model. Moreover, for each dichotomous outcome, the pooled risk ratio (RR) with its 95% CI was calculated using the Der Simonian–Laird random-effect model.

Statistical heterogeneity among studies was evaluated by the Chi-square test (Cochrane Q test). An I^2^ test for heterogeneity was then determined according to the equation: I^2^ = (Q − df)/Q × 100%. A Chi-square *p*-value less than 0.05 and I-square values ≥ 50% were considered to represent significant heterogeneity. Revman V 5.3 software (The Cochrane Collaboration, London, UK) was used for all statistical analyses.

## 3. Results

### 3.1. Literature Search Results

A total of 1518 citations were considered, and 639 citations were screened following the removal of duplicates. Subsequently, a total of 30 articles were eligible for full-text screening following title and abstract screening. Finally, four studies met the inclusion criteria for this systematic review and meta-analysis [13,14,15,16]. The detailed study selection process is shown in the PRISMA diagram (Figure 1).

### 3.2. Study Characteristics

A total of four RCTs comprising 1981 patients were included. Among these participants, 1349 were male. The average age of the cohort was 55 years, with a mean BMI of 25 kg/m^2^. Notably, the follow-up durations across all four RCTs were 12 weeks, except for Rakugi et al. (2019) [15], which extended from 18 to 52 weeks. Importantly, all enrolled patients had HTN grade I/II. Detailed baseline and summary characteristics are provided in Table 1.

### 3.3. Quality Assessment

In our included RCTs, two studies conducted by Ito et al. and Ito et al. [13,14] showed a lower risk of bias in all domains, while Rakugi et al. [15] and Kario et al. [16] were judged as having some concerns, and both of them reported some concerns in the randomization process. Additionally, Kario et al. [16] showed some concerns in the measurement of the outcome process, and Rakugi et al. [15] reported bias due to deviation from the intended intervention. Detailed ROB2 domains are provided in Figure 2.

### 3.4. Primary Efficacy Outcomes

#### Sitting BP

For sitting DBP, the 1.25 mg subgroup (four studies; 704 patients in the intervention group vs. 530 in the control group) had a reduction of 0.97 mmHg that was not statistically significant (95% CI −3.24 to 1.30; *p* = 0.40; I^2^ = 83%, Chi^2^-*p* = 0.0004), while the 5 mg subgroup (two studies; 425 vs. 418) demonstrated a borderline 4.22 mmHg reduction (95% CI −8.72 to 0.29; *p* = 0.07; I^2^ = 94%, Chi^2^-*p* = 0.0001). Overall (1129 vs. 948), there was a modest but significant 2.08 mmHg reduction (95% CI −4.07 to −0.08; *p* = 0.04; I^2^ = 89%, Chi^2^-*p* < 0.00001) (Figure 3).

For sitting SBP, the 1.25 mg subgroup (four studies; 704 patients in the intervention group vs. 530 in the control group) showed a non-significant 2.73 mmHg reduction (95% CI −6.04 to 0.59; *p* = 0.11; I^2^ = 74%, Chi^2^-*p* = 0.008), while the 5 mg subgroup (two studies; 425 vs. 418) achieved a borderline 9.07 mmHg reduction (95% CI −17.69 to −0.45; *p* = 0.04; I^2^ = 95%, Chi^2^-*p* < 0.00001). Pooled data (1129 vs. 948) confirmed a significant 5.01 mmHg overall reduction (95% CI −8.58 to −1.43; *p* = 0.006; I^2^ = 90%, Chi^2^-*p* < 0.00001) (Figure 4).

### 3.5. Secondary Efficacy Outcomes

#### 3.5.1. Twenty-Four h BP

For 24 h DBP, the 1.25 mg subgroup (three studies; 659 patients in the intervention group vs. 480 in the controls) showed a non-significant 0.38 mmHg reduction (95% CI −4.31 to 3.55; *p* = 0.85) with marked heterogeneity (I^2^ = 94%, Chi^2^-*p* < 0.00001). In contrast, the 5 mg subgroup (two studies; 425 vs. 416) yielded a significant 5.98 mmHg reduction (95% CI −11.76 to −0.20; *p* = 0.04) with substantial heterogeneity (I^2^ = 96%, Chi^2^-*p* < 0.00001). Overall analysis (1084 vs. 896), Esaxerenone was associated with a non-significant 2.62 mmHg reduction (95% CI −5.73 to 0.49; *p* = 0.10) and continued high heterogeneity (I^2^ = 96%, Chi^2^-*p* < 0.00001) (Figure 5).

For 24 h SBP, the 1.25 mg subgroup (three studies; 659 vs. 480) demonstrated a non-significant 2.08 mmHg reduction (95% CI −10.53 to 6.37; *p* = 0.63) with notable heterogeneity (I^2^ = 91%, Chi^2^-*p* = 0.0001), whereas the 5 mg subgroup (two studies; 425 vs. 416) achieved an 11.87 mmHg reduction (95% CI −22.41 to −1.34; *p* = 0.03) with similar heterogeneity (I^2^ = 93%, Chi^2^-*p* = 0.0001). When data from both doses were pooled (1084 vs. 896), the analysis indicated a non-significant 6.05 mmHg reduction (95% CI −13.48 to 1.39; *p* = 0.11) with high heterogeneity (I^2^ = 94%, Chi^2^-*p* < 0.00001) (Figure 6).

#### 3.5.2. Target BP Achievement

Assessment of target BP achievement indicated a non-significant 23% increase (RR 1.23, 95% CI 0.94 to 1.60; *p* = 0.12; I^2^ = 41%, Chi^2^-*p* = 0.16) with 1.25 mg (four studies; 704 vs. 528), whereas the 5 mg subgroup (two studies; 425 vs. 416) achieved a twofold increase (RR 2.05, 95% CI 1.03 to 4.08; *p* = 0.04; I^2^ = 85%, Chi^2^-*p* = 0.01). The pooled analysis (1129 vs. 944) revealed a significant overall 48% increase (RR 1.48, 95% CI 1.11 to 1.96; *p* = 0.007; I^2^ = 72%, Chi^2^-*p* = 0.003) (Figure A1).

#### 3.5.3. BP During Different Times Measurements

Subgroup analyses of daytime SBP showed a borderline 4.34 mmHg reduction (95% CI −9.00 to 0.31; *p* = 0.07; I^2^ = 71%, Chi^2^-*p* = 0.03) with 1.25 mg (three studies; 459 vs. 466) versus a significant 12.41 mmHg reduction (95% CI −20.78 to −4.03; *p* = 0.004; I^2^ = 91%, Chi^2^-*p* = 0.0006) with 5 mg (two studies; 425 vs. 418). Overall (884 vs. 884), esaxerenone reduced SBP by 7.82 mmHg (95% CI −12.91 to −2.72; *p* = 0.003) with considerable heterogeneity (I^2^ = 91%, Chi^2^-*p* < 0.00001) (Figure A2).

Analyses of morning SBP revealed a significant 5.96 mmHg reduction (95% CI −8.96 to −2.96; *p* < 0.0001; I^2^ = 42%, Chi^2^-*p* = 0.18) at the 1.25 mg dose (three studies; 459 vs. 466), whereas the 5 mg subgroup (two studies; 425 vs. 418) showed a larger 14.81 mmHg reduction (95% CI −25.98 to −3.65; *p* = 0.009; I^2^ = 94%, Chi^2^-*p* < 0.0001). Overall (884 vs. 884), the pooled effect was a significant 9.45 mmHg reduction (95% CI −14.20 to −4.71; *p* < 0.0001) with high heterogeneity (I^2^ = 89%, Chi^2^-*p* < 0.00001) (Figure A3).

For nocturnal SBP, the 1.25 mg subgroup (three studies; 459 vs. 466) showed a 7.81 mmHg reduction (95% CI −9.85 to −5.76; *p* < 0.00001) with low heterogeneity (I^2^ = 13%, Chi^2^-*p* = 0.32), and the 5 mg subgroup (two studies; 425 vs. 418) demonstrated a more marked 14.19 mmHg reduction (95% CI −18.43 to −9.95; *p* < 0.00001) with moderate heterogeneity (I^2^ = 68%, Chi^2^-*p* = 0.08). Overall (884 vs. 884), esaxerenone significantly reduced nocturnal SBP by 10.62 mmHg (95% CI −14.01 to −7.23; *p* < 0.00001; I^2^ = 82%, Chi^2^-*p* = 0.0002) (Figure A4).

### 3.6. Safety Outcomes

Drug related adverse events revealed a non-significant difference with the 1.25 mg dose (RR 1.19, 95% CI 0.84 to 1.69; *p* = 0.32; I^2^ = 0%, Chi^2^-*p* = 0.68) and an increase of 76%, which was statistically significant with the 5 mg dose (RR 1.76, 95% CI 1.09 to 2.84; *p* = 0.02; I^2^ = 0%, Chi^2^-*p* = 0.64). Overall, there was a 38% increase in adverse events (RR 1.38, 95% CI 1.04 to 1.82; *p* = 0.03; I^2^ = 0%, Chi^2^-*p* = 0.63) (Figure A5).

Over all adverse events were not elevated at either the 1.25 mg dose (RR 1.13, 95% CI 0.63 to 1.99; *p* = 0.69) or the 5 mg dose (RR 1.97, 95% CI 0.68 to 5.70; *p* = 0.21), yielding a non-significant overall 29% increase (RR 1.29, 95% CI 0.78 to 2.13; *p* = 0.32; I^2^ = 0%, Chi^2^-*p* = 0.55). Discontinuations did not yield a significant difference (RR 2.52, 95% CI 0.87 to 7.26; *p* = 0.09; I^2^ = 0%, Chi^2^-*p* = 0.86). Finally, there was no significant difference in the rate of all adverse events (RR 1.02, 95% CI 0.93 to 1.13; *p* = 0.66; I^2^ = 38%, Chi^2^-*p* = 0.15) (Figure A6).

For potassium level increases, the risk ratio of 1.29 [0.78, 2.13] is not statistically significant (*p* = 0.32). For serum uric acid increases, the risk ratio of 2.52 [0.87, 7.26] trends higher but also fails to reach statistical significance (*p* = 0.09) (Figure A7 and Figure A8).

## 4. Discussion

Our meta-analysis of four randomized controlled trials involving 1981 patients evaluated the efficacy and safety of Esaxerenone for hypertension at 1.25 mg and 5 mg doses. Findings revealed significant dose-dependent blood pressure reductions, with the 5 mg dose demonstrating superior efficacy compared with the 1.25 mg dose across multiple measurements. Overall, esaxerenone reduced sitting SBP by 5.01 mmHg (*p* = 0.006) and sitting DBP by 2.08 mmHg (*p* = 0.04), while increasing target BP achievement by 48% (*p* = 0.007). The medication showed particularly strong effects on nocturnal blood pressure, with reductions of 7.81 mmHg and 14.19 mmHg for the 1.25 mg and 5 mg doses, respectively. Regarding safety, there was a 38% increase in adverse events (*p* = 0.03), primarily with the 5 mg dose, but no significant increase in serious adverse events or overall adverse event rates, suggesting a favorable risk-benefit profile, especially at the lower dose.

Our findings corroborate Ito et al.’s multicenter study (*n* = 1001) demonstrating esaxerenone dose-dependent efficacy in Japanese hypertensive patients, with 2.5 mg/day achieving non-inferiority to eplerenone (50 mg/day) and 5 mg/day showing superior antihypertensive effects [14]. Concurrently, Rakugi et al.’s phase 3 trial (*n* = 594) revealed clinically meaningful sitting SBP reductions with esaxerenone (whether as monotherapy or combination therapy), progressively increasing from 15–17 mmHg at week 12 to 21–24 mmHg at week 52 [15,21].

Plasma aldosterone concentration (PAC) and plasma renin activity (PRA) were examined as markers to assess the effectiveness of esaxerenone in blocking mineralocorticoid receptors, as these indicators are raised by MR blockade. Increasing PAC and PRA were observed with 2.5 mg/day and 5 mg/day doses of esaxerenone during treatment, indicating that inhibiting aldosterone receptors is responsible for esaxerenone blood pressure-lowering effects [14,15]. Furthermore, compared with eplerenone, esaxerenone exhibited higher levels of PAC and PRA, suggesting that esaxerenone may have a more substantial impact on renal tubules and more potent inhibition of the M.R. in other tissues than eplerenone [14]. The consistent elevation of PAC and PRA across studies suggests that esaxerenone’s mechanism of action likely involves effective mineralocorticoid receptor blockade, distinguishing it from drugs targeting other pathways.

In several prior studies, consistent with our findings, esaxerenone demonstrated sustained antihypertensive effects throughout the 24 h dosing period. These effects are likely due to its distinct characteristics as a novel MR blocker, including potent MR inhibition and a prolonged half-life compared with other agents in this class, possibly due to its non-steroidal structure [12,13,14,15]. The favorable effects of esaxerenone therapy demonstrated in these studies were more evidenced by the gradual and sustained decreases in natriuretic peptide levels throughout the treatment duration, remaining within normal levels and consistent with those observed with other medications targeting the M.R. [15,21]. All these findings about esaxerenone are expected to have substantial implications for clinical practice.

Esaxerenone has been shown to improve dipping patterns and reduce riser types [22]. This effect, combined with its direct nocturnal blood pressure reduction [16], suggests a potential role in lowering risks linked to abnormal 24 h BP profiles. This could be attributed to several potential mechanisms; the reduction in nighttime blood pressure is correlated with the circulating blood volume. Consequently, the robust M.R. inhibitory effect of esaxerenone might reduce nighttime blood pressure by diminishing fluid volume by inhibiting sodium reabsorption [11]. However, there are also theoretical associations between M.R. antagonism, aldosterone, resistant hypertension, and sleep apnea [23,24], which may be relevant to our findings.

In cases where risk factors for nocturnal hypertension, like sympathetic hyperactivity and RAS activation, are present, resulting in elevated plasma aldosterone levels and M.R. activation leading to salt and water retention [21], MRBs are regarded as an ideal class of antihypertensive drugs capable of managing nocturnal hypertension. MRBs such as spironolactone or eplerenone have been reported to manage nocturnal hypertension [25,26,27,28]. Still, the substantially stronger affinities of esaxerenone for M.R. binding and its longer half-life suggest that it may have a more sustained and potent impact on lowering nighttime blood pressure than other MRBs [28].

Patient characteristics influence the effectiveness of esaxerenone in lowering blood pressure, the subgroup analysis of the ESAX-HTN study revealed that in individuals with essential hypertension, reduction of in-office blood pressure was slightly more substantial in female patients, those aged 65 years or older, individuals with a body mass index below 25 kg/m^2^, and those with lower baseline levels of PAC and PRA (less than 120 pg/mL and 1.0 ng/mL/h, respectively) [14]. This finding was consistent with a subgroup analysis of the EARLY-NH study, which showed a similar trend to the previous results [19]. A more pronounced reduction was observed in females, yet the reasons for these differences between genders have not been entirely determined. However, recent studies suggested that these differences may result from greater salt sensitivity and more SBP, aldosterone, and renal plasma flow response to angiotensin II in women [29,30].

Safety analyses revealed no significant differences between esaxerenone and standard therapy/placebo regarding adverse events and potassium elevation. Ito et al. reported comparable serum potassium elevation risks versus placebo [14]. Notably, esaxerenone demonstrated absence of sex hormone-related adverse events, consistent with non-clinical studies showing no agonist/antagonist activity on glucocorticoid, progesterone, or androgen receptors [9,31]. This improved risk-benefit profile stems from enhanced antihypertensive efficacy with safety equivalent to eplerenone [14]. Conversely, esaxerenone exhibited significantly higher drug-related adverse event risk compared with standard therapy. Two events prompted treatment discontinuation: altered consciousness (potentially attributable to concurrent cerebral hemorrhage following head trauma) and generalized rash. While the rash’s temporal relationship to treatment suggested causality, definitive attribution to esaxerenone remains unestablished [14].

Our study constitutes the most comprehensive systematic review and meta-analysis comparing esaxerenone against conventional therapies and placebo in essential hypertension management. In contrast to the previous meta-analysis by Sun et al. [17], which was limited to only two RCTs without a thorough evaluation of all BP parameters, our analysis distinguishes itself by exclusively incorporating all up-to-date clinical trials, thereby enhancing methodological rigor. Furthermore, we employed comprehensive statistical frameworks, including dosage-stratified analyses, to identify optimal therapeutic parameters.

Nevertheless, certain methodological constraints warrant acknowledgment. The relatively constrained study pool and participant numbers potentially diminished statistical power for detecting meaningful clinical differences. Future investigations should address numerous confounding variables, particularly regarding baseline patient characteristics, antihypertensive medication history, both as monotherapies and combination regimens, and type of hypertension (resistant status). Such refinements would substantially strengthen the evidentiary foundation for clinical recommendations regarding this therapeutic agent. Another limitation is that all trials included in the current meta-analysis were conducted exclusively in Japan. Consequently, the external validity and generalizability of the findings to populations of other races or ethnicities may be limited. However, it should be noted that these represent all RCTs conducted to date on this topic.

The present meta-analysis highlights the necessity of standardizing reporting in future research on the utility of esaxerenone for treating essential hypertension. Additionally, determining the optimal dosage for treatment is crucial for optimizing its efficacy and safety outcomes. All our findings regarding esaxerenone are expected to have substantial implications for clinical practice, suggesting the potential for more robust blood pressure control. However, monitoring serum potassium levels is crucial for patients over 80 years old and those with moderate renal dysfunction due to reported adverse events associated with elevated potassium levels from the drug.

## 5. Conclusions

Esaxerenone may be a valuable addition to the antihypertensive armamentarium, particularly in patients with nocturnal hypertension or intolerance to steroidal MR antagonists. Our meta-analysis demonstrates that esaxerenone provides significant dose-dependent blood pressure reductions in hypertensive patients, with the 5 mg dose showing superior efficacy. The medication effectively reduces sitting SBP and DBP while increasing target BP achievement, with particularly notable effects on nocturnal blood pressure. Although adverse events were more frequent at the higher dose, the overall safety profile remained acceptable. Future studies should aim to determine optimal dosing strategies for different patient populations, with special attention to elderly patients and those with renal dysfunction, where potassium monitoring is essential. Long-term efficacy and safety outcomes should also be prioritized to better inform clinical practice.

## Figures and Tables

**Figure 1 jcm-14-05663-f001:**
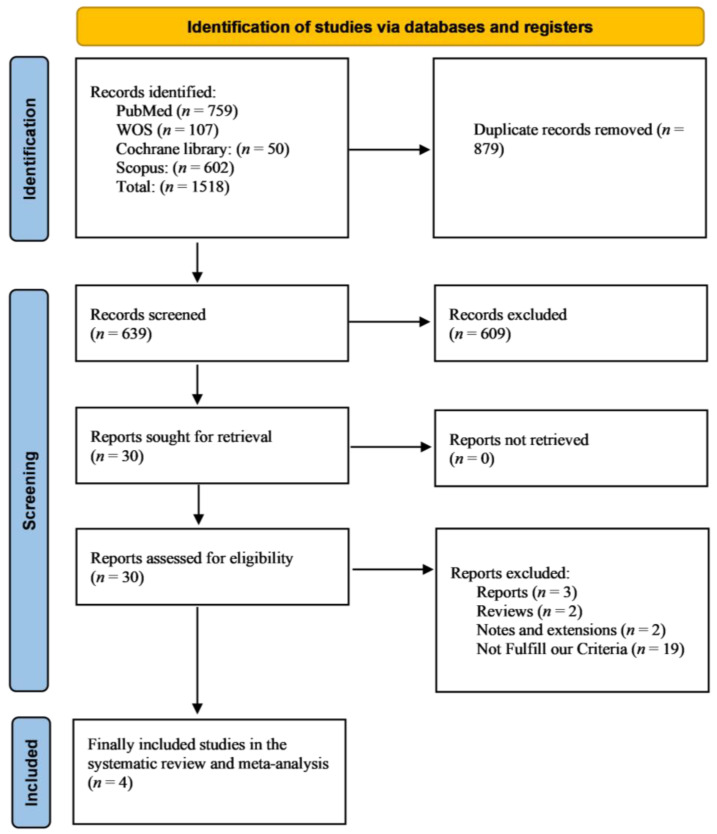
Presents the PRISMA flow diagram, illustrating the study selection process.

**Figure 2 jcm-14-05663-f002:**
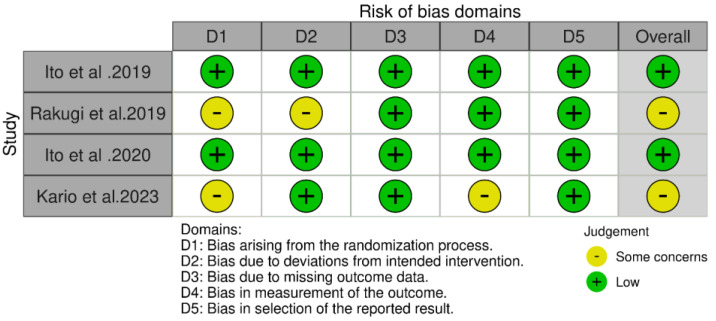
Provides ROB1 summary for our four included studies [13,14,15,16].

**Figure 3 jcm-14-05663-f003:**
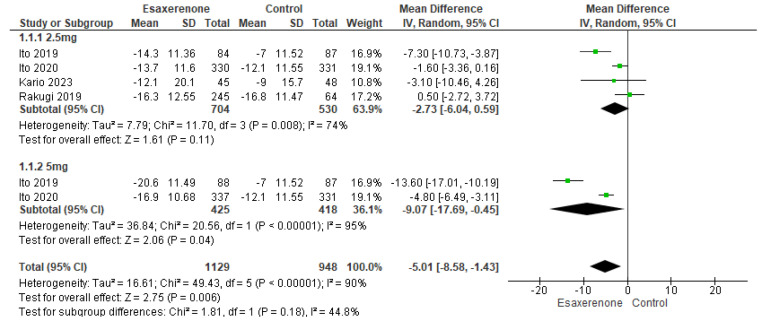
Represents the forest plot for sitting SBP [13,14,15,16].

**Figure 4 jcm-14-05663-f004:**
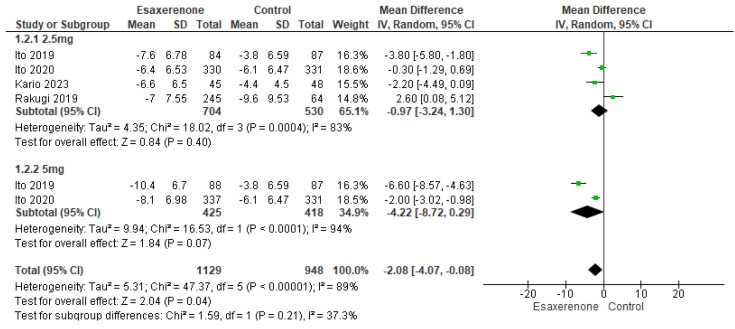
Represents the forest plot for Sitting DBP [13,14,15,16].

**Figure 5 jcm-14-05663-f005:**
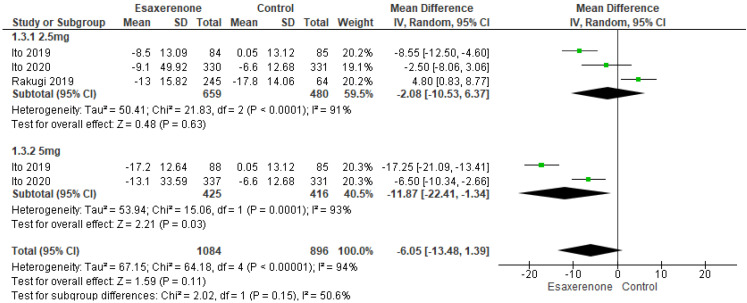
Depicts a forest plot for 24 h systolic BP [13,14,15,16].

**Figure 6 jcm-14-05663-f006:**
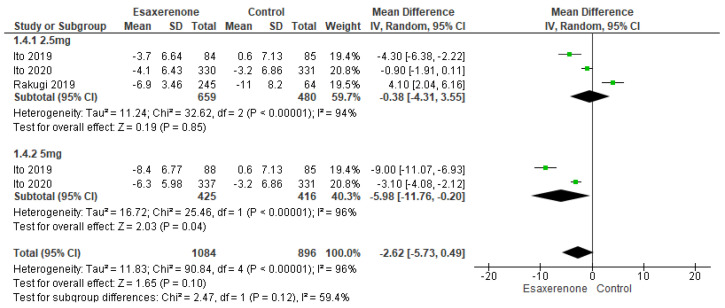
Depicts a forest plot for 24 h diastolic BP [13,14,15,16].

**Table 1 jcm-14-05663-t001:** Baseline characteristics and summary of the included studies.

Study ID	Study Arms (%)	Site	Trial Registration	Age, (Mean ± SD) y	Male, n (%)	BMI, (Mean ± SD) (kg/m^2^)	Follow-Up Duration (Months)	Daily Dose (mg)	Hypertension Grade, n (%)	Prior Treatment of HTN, n (%)	Duration of HTN, Months	Past History, n (%)	Inclusion Criteria	Primary Endpoints	Conclusion
Ito, 2019 [13]	Esaxerenone (1.25, 2.5, or 5 mg/day), placebo, or eplerenone (50–100 mg/day)	Japan	NR	57.0 ± 9.3	295 (69.7)	25.5 ± 4.0	12 weeks	Esaxerenone (1.25, 2.5, or 5 mg/day), eplerenone (50–100 mg/day)	Grade I or II	221 (52.2)	NR	Diabetes 58 (13.7), LDL cholesterol (mg/dL) 129.3 ± 31.8	Inclusion criteria: aged ≥ 20 years at time of informed consent; sitting systolic BP (SBP) of ≥140 to <180 mmHg and diastolic BP (DBP) ≥90 to <110 mmHg; and 24 h BP by ambulatory BP monitoring (ABPM) of ≥130/80 mmHg.	Change from baseline in sitting BP (SBP and DBP) at the end of the treatment period, defined as the average sitting BP of week 10 and week 12 after last observation carried forward (LOCF) imputation of missing values.	The incidence of adverse events was similar in all treatment groups. Serum K+ levels initially increased in proportion to esaxerenone dose but were stable from week 2 until week 12. Plasma esaxerenone concentration increased in proportion to the dose. In conclusion, esaxerenone is an effective and tolerable treatment option for patients with essential hypertension.
Ito, 2020 [14]	Esaxerenone 2.5 or 5 mg/day or eplerenone 50 mg/day	Japan	NCT02890173	55.5 ± 9.6	721 (72.2)	25.7 ± 4.1	12 weeks	Esaxerenone 2.5 or 5 mg/day or eplerenone 50 mg/day	Grade I: 454 (45.5) or II: 544 (54.5)	514 (51.5)	NR	Comorbid type 2 diabetes mellitus 156 (15.6), Triglycerides, mg/dL 138.1 ± 115.1	Included patients who provided informed consent, received ≥1 dose of study drug, and had at least one efficacy measurement. The per-protocol set (PPS) included patients who completed study treatments without major protocol deviations or missing primary end point data and had study drug compliance of ≥75%. The safety analysis set included all those who received ≥1 dose of the study drug.	Changes in sitting SBP and DBP from baseline until the end of treatment (defined as mean BP calculated using values from weeks 10 and 12).	These results indicate that esaxerenone is an effective and well-tolerated MR blocker in Japanese patients with essential hypertension, with BP-lowering activity at least equivalent to eplerenone.
Kario, 2023 [16]	Angiotensin receptor blocker, calcium-channel blocker	Japan	NR	67.6 ± 11.6	47 (50.5)	25.5 ± 4.3	12 weeks	The starting dose was 2.5 mg, which could be titrated to 5 mg if the response was inadequate	NR	NR	140.2 ± 122.1	Type 2 DM 33 (35.5), diabetic retinopathy 8 (8.6), dyslipidemia 52 (55.9), hyperuricemia 17 (18.3), heart failure 1 (1.1), smoking 14 (15.1)	Received at least one dose of esaxerenone, and had at least one efficacy endpoint evaluation. The per-protocol set (PPS) was defined as FAS patients who adhered to the package insert of esaxerenone.	The change in nighttime home SBP and DBP measured with the brachial device from baseline to the end of treatment (EOT).	Esaxerenone was effective in lowering nighttime home BP as well as morning and bedtime home BP and office BP, was safe, and showed organ-protective effects in patients with uncontrolled nocturnal hypertension. Caution is warranted regarding elevated serum potassium levels.
Rakugi, 2019 [15]	Patients received esaxerenone monotherapy or esaxerenone in combination with a CCB or RAS inhibitor.	Japan	NR	56.2 ± 9.2	286 (77.7)	25.7 ± 3.6	28, 52 weeks	Patients were treated with esaxerenone starting at 2.5 mg/day, increasing to 5 mg/day if required to achieve blood pressure (BP) targets.	Grade I: 176 (47.8) or II: 192 (52.2)	244 (66.3)	NR	Diabetes 67 (18.2)	Received the study drug at least once, and had efficacy endpoint data measured at least once during the treatment period.	The primary endpoint was a change from baseline in sitting BP	Esaxerenone was also well-tolerated with a rate of hyperkalemia at 5.4% (serum potassium ≥ 5.5 mEq/L), indicating a good safety profile for treatment over the long term or in combination with a CCB or RAS inhibitor. In conclusion, esaxerenone may be a promising treatment option for patients with hypertension.

## Data Availability

The original contributions presented in this study are included within the article. Further inquiries can be directed to the corresponding author(s).

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
