# Peer review of "Efficacy and Safety of Esaxerenone for Essential Hypertension: A Systematic Review and Meta-Analysis of Randomized Controlled Trials†"

_jcm, 2025, doi:10.3390/jcm14165663_

Round 1
Reviewer 1 Report
Comments and Suggestions for Authors
The article “Efficacy and Safety of Esaxerenone for Essential Hypertension: A systematic review and meta-analysis of randomized controlled trial” addresses a relevant clinical question about the use of esaxerenone in essential hypertension, through a systematic review with meta-analysis. It is well structured, with clear methodology, detailed results and well-founded discussion. However, some grammatical corrections, scientific style and clarification of some sections are necessary.
Title
"Efficacy and Safety of Esaxerenone for Essential Hypertension: A systematic review and meta-analysis of randomized controlled trials" is suggested as "...randomized controlled trials"
Abstract
Line 17 "shows promising results for treating..." is suggested as "has shown promising results in the treatment of..."
Line 34 "Esaxerenone showed to be safe..." is suggested as "Esaxerenone was shown to be safe..."
Line 25 "Safety measures as adverse events..." is suggested as "Safety outcomes such as adverse events..."
Introduction
Some sentences are long and can be rewritten for greater flow.
It is suggested that a final paragraph be inserted with a clear statement of the objective. It is suggested as:
"Thus, we conducted a systematic review and meta-analysis of RCTs to assess the efficacy and safety of esaxerenone in adult patients with essential hypertension." Line 40 “uncontrolled blood pressure despite proper management...” suggests “despite optimal management...”
Line 57 “to control different sittings of BP” suggests “to control blood pressure under different conditions”
Methods
“We included only RCTs that included patients...” suggests “included only RCTs enrolling...”
Line 72 “references in a two-step fashion” suggests “in a two-step process”
Line 76 “reported outcomes of interest of patients allocated to...” suggests “reported outcomes of interest in patients allocated to...”
Line 105 “I2= (Q-df/Q) x100%” suggests formatting correctly as a mathematical formula: I² = (Q - df) / Q × 100%
Suggested to correct inconsistent capitalization (“Literature Search”, “Eligibility criteria”) — follow JCM AMA Style. It is important to highlight the GRADE instrument, if it was used, as it was not mentioned.
Results
Line 132 “non-significant 0.97 mmHg reduction” suggests “a reduction of 0.97 mmHg that was not statistically significant”
Line 182 “pronounced 14.19 mmHg reduction” suggests “a marked 14.19 mmHg reduction”
Line 190 “a significant 76% increase” suggests “an increase of 76%, which was statistically significant”
Line 201 “is not statistically significant (P = 0.32)” suggests “was not statistically significant (p = 0.32)” – lowercase for p
I suggest explicitly mentioning the studies included in the first sentence (authors + year).
I suggest including absolute pre- and post-treatment BP values whenever possible for greater clinical clarity. There is repeated use of the expression “non-significant”: consider alternating it with “not statistically significant” or “did not reach significance”.
Many figures and tables are referenced, but the current layout of the manuscript needs to improve the coherence between text and figures/supplements.
Discussion
Well structured and with extensive contextualization in the literature.
“...confirming esaxerenone mechanism of action...” “esaxerenone’s mechanism” is suggested.
“...diminishing fluid volume by inhibiting sodium reabsorption [9]. However, there are also theoretical associations...” is suggested to be separated into two distinct paragraphs for clarity.
Line 240–241 “as these indicators are raised by M.R. blockade” suggests “since these markers are elevated by MR blockade”
Line 248 “esaxerenone mechanism of action” suggests “esaxerenone’s mechanism of action”
Line 261–262 “further supports its potential role in mitigating the risks associated with abnormal 24-hour BP profiles...” suggests dividing the sentence into two for better readability.
Line 284–286 “...increased salt sensitivity and more SBP, aldosterone, and renal plasma flow response...” suggests “...greater salt sensitivity, and a more pronounced response in SBP, aldosterone, and renal plasma flow...”
Suggests emphasizing the clinical implication of nocturnal efficacy, which is a differential.
Conclusion
Clear and in line with the findings. You can reinforce:
“Esaxerenone may be a valuable addition to the antihypertensive armamentarium, particularly in patients with nocturnal hypertension or intolerance to steroidal MR antagonists.”
Line 321 “dose-dependent” correct the line break: “dose-dependent”
Line 325 “Despite a modest increase in adverse events primarily associated with the higher dose...” it is suggested “Although adverse events were more frequent at the higher dose, the overall safety profile remained acceptable.”
References
It is suggested to standardize the use of authors' names in citations (e.g.: “Ito et al.” vs. “Ito et al., 2020”).
Minor revision: the article is solid and relevant, but requires linguistic adjustments, formal corrections, and adjustments to the scientific style to fully meet the standards
Author Response
Dear Editor and reviewers,
Thank you for considering our paper, and all these valuable comments which will, for sure, improve our paper.
Reviewer 1
|
Reviewer Comment |
Author Response |
Exact Edit Location |
|
1. The manuscript is well structured, but several grammatical, stylistic, and formatting improvements are suggested throughout the title, abstract, introduction, methods, results, discussion, conclusion, and references. 2. Standardization of references (e.g., “Ito et al.” vs. “Ito et al., 2020”) is also recommended. |
1. We sincerely thank the reviewer for their careful reading and constructive suggestions. We have thoroughly revised the manuscript to address all grammar, syntax, and scientific style recommendations. Specific phrasing throughout the abstract, results, and discussion has been improved for clarity and scientific tone, as suggested. 2. Regarding the reference style: while we follow the Vancouver style as required, we occasionally included the publication year after the author name (e.g., “Ito et al., 2020”) only when needed to differentiate between multiple studies by the same author (e.g., Ito et al., 2020 vs. Ito et al., 2019). This was done solely to avoid confusion in instances where the same author conducted more than one study included in our analysis. All references have now been carefully reviewed and standardized. We appreciate your thoughtful feedback. |
All changes have been highlighted yellow on the whole manuscript |
Reviewer 2
|
Reviewer Comment |
Author Response |
Exact Edit Location |
|
The Introduction is too short. Authors are encouraged to provide more information, including epidemiological data and pathophysiology of resistant hypertension. References are limited and should be strengthened. |
1. We sincerely thank the reviewer for this valuable feedback. In response, we have thoroughly revised and expanded the Introduction. 2. We incorporated current epidemiological data and provided a concise overview of the pathophysiological mechanisms underlying resistant hypertension. 3. Additionally, we have significantly enriched the reference list with recent and relevant studies to enhance the scientific context and credibility of the background section. We hope these improvements meet the reviewer’s expectations. |
All changes have been highlighted yellow. Introduction from line 43 to line 49 and from line 63 to line 67 |
Reviewer 3
|
Reviewer Comment |
Author Response |
Exact Edit Location |
|
1. The reviewer identified a previously published meta-analysis (Sun et al., Journal of Human Hypertension, 2024). The authors should acknowledge this study in the Introduction and Discussion and clarify how their work differs. |
We thank the reviewer for this important point. The mentioned meta-analysis by Sun et al. (2024) has now been cited in both the Introduction and Discussion. As clarified in our revised Discussion section, that meta-analysis has significant limitations. It relies solely on two RCTs, lacks stratified or subgroup analyses, and does not explore detailed blood pressure parameters as we have done in our current work. |
Introduction: |
|
2. All included trials were conducted in Japan. Therefore, the generalizability of the findings may be limited. |
We agree with the reviewer and have now explicitly addressed this limitation in the revised Discussion. |
Discussion |
|
3. The Introduction discusses resistant hypertension, but the included studies do not specifically target this population. This may cause confusion. |
Thank you for your comment. While the available trials of Esaxerenone primarily enrolled patients with essential hypertension not necessarily resistant, I began the introduction with resistant hypertension because this reflects the main clinical setting where mineralocorticoid receptor antagonists (MRAs) are typically used. As outlined in ACC/AHA and ESC/ESH guidelines, MRAs are recommended when blood pressure remains uncontrolled despite optimal use of three or more antihypertensive agents, including a diuretic—characterizing resistant hypertension.
Importantly, although the trials did not strictly limit inclusion to resistant hypertension, many included patients who likely met that definition and referenced the potential role of Esaxerenone in this population, consistent with the recognized role of MRAs in current guidelines. However, due to the small number of existing studies (only four), strict stratification was not applied.
We have acknowledged this in our limitations section, noting that future research should include clearer subgrouping and stratification to better define Esaxerenone’s efficacy specifically in resistant hypertension. Nonetheless, as a meta-analysis, our work includes all currently available evidence to date.
|
Lines 322-330 |
|
4. Did the authors assess publication bias? If not, a justification may be provided. |
Due to the limited number of included studies, formal publication bias assessment (e.g., funnel plot or Egger’s test) was not performed, in accordance with Cochrane guidelines, which caution against such analyses with fewer than 10 studies. This rationale has been added to the Methods section. |
Methods |
|
5. It is unclear whether concomitant antihypertensive medications affected the magnitude of blood pressure reduction. Subgroup analysis or discussion is encouraged. |
We appreciate this observation. While several included studies allowed or reported background antihypertensive use, detailed stratified outcome data were not consistently provided. Therefore, we could not perform a formal subgroup analysis. However, we have added this point to the Discussion and acknowledged it as a limitation of the available evidence. |
Discussion |
|
6. The forest plots for blood pressure outcomes have overly broad x-axes, making interpretation difficult. |
Thank you for this suggestion. We have updated the forest plots for all blood pressure outcomes, particularly the first two plots, using improved scaling for clearer visual interpretation. The revised figures are included in the updated manuscript. |
Page 7 |
We want to give warm thanks to the reviewers for their valuable comments as we really appreciate all these comments as it improved our paper a lot.
Best regards,

Reviewer 2 Report
Comments and Suggestions for Authors
Dear Authors, Your interesting paper focuses on an exciting field of research so it's really worthy of interesting. On my opinion, introduction is too short. Authors are encouraged to provide further informations about the topic wit epidemyologycal data and pathophysiology of the mechanisms underlying reistant hypertension. Literature is rich of papers and review about this topic. References are poor. Please empower them.
No odds about the further sections.
Author Response

(The authors gave the same response as above.)

Reviewer 3 Report
Comments and Suggestions for Authors
The authors conducted a systematic review and meta-analysis to evaluate the antihypertensive efficacy and safety of esaxerenone in patients with essential hypertension. The findings support the utility of esaxerenone as a potential treatment option for blood pressure control. However, the reviewer has several concerns that should be addressed to enhance the clarity and scientific rigor of the manuscript.
- The reviewer has two major concerns. First, the reviewer identified a previously published meta-analysis addressing the same topic: the efficacy and safety of esaxerenone (Sun et al., Journal of Human Hypertension, 2024;38:102–109). The authors should acknowledge this study in the Introduction and Discussion, and briefly clarify how their current work differs in methodology, scope, included trials, or key findings.
- Second, all included trials in the current meta-analysis were conducted in Japan. Therefore, the external validity and generalizability of the findings to other race/ethnicity may be limited. This limitation should be explicitly discussed.
- The Introduction discusses resistant hypertension; however, the present meta-analysis does not appear to include studies that specifically target resistant hypertension. This discrepancy may confuse readers. The authors are encouraged to revise the Introduction to clarify the scope of the study population and explain the relevance of resistant hypertension in the context of this analysis.
- Did the authors assess publication bias? If not, a justification may be provided, especially considering the small number of included studies, which may limit the reliability of formal bias testing.
- It is unclear whether the presence or absence of concomitant antihypertensive medications affected the magnitude of blood pressure reduction. If data are available, subgroup or sensitivity analyses may be considered or discussed.
- In the forest plots of blood pressure outcomes, the x-axis (horizontal scale) appears overly broad, making it difficult to interpret differences between studies.
Author Response

(The authors gave the same response as above.)

Round 2
Reviewer 3 Report
Comments and Suggestions for Authors
The authors have carefully addressed the reviewer’s previous comments. The reviewer thanks the authors for their diligent revisions.
The reviewer has one minor suggestion: please consider adding a statement in the Abstract noting that all included trials were conducted in Japan.
Author Response
Thank you for your thoughtful feedback and kind acknowledgment. We appreciate the reviewer’s suggestion and have added a statement to the Abstract noting that all included trials were conducted in Japan. This addition has been highlighted in green for ease of reference.
